Prediction of antigenic peptides of SARS- CoV-2 pathogen using machine learning

Bukhari Syed Nisar Hussain 1 nisar.bukhari@gmail.com
http://orcid.org/0000-0001-6589-4904 Ogudo Kingsley A. 2
1 National Institute of Electronics and Information Technology (NIELIT) , Srinagar, J&K , India
2 Department of Electrical & Electronics Engineering Faculty of Engineering and the Built Environment, University of Johannesburg , Johannesburg , South Africa
Zhou Jiayan
Electronic publication date: 2024 Oct 10
Publication date: 2024
Volume: 10
Electronic Location ID: e2319
Received 2024 May 1; Accepted 2024 Aug 20
Copyright: © 2024 Bukhari and Ogudo
Copyright year: 2024
Copyright holder: Bukhari and Ogudo
License: This is an open access article distributed under the terms of the Creative Commons Attribution License, which permits unrestricted use, distribution, reproduction and adaptation in any medium and for any purpose provided that it is properly attributed. For attribution, the original author(s), title, publication source (PeerJ Computer Science) and either DOI or URL of the article must be cited.
License URL: https://creativecommons.org/licenses/by/4.0/

Keywords: Antigenic peptide, Machine learning, XGBoost, Epitope-based vaccine, SARS-CoV-2, COVID-19, T-cell epitope

Funding: University of Johannesburg’s University Research Committee (URC) Department of Electrical and Electronic Engineering Technology University of Johannesburg Library Research Funds (UJ) This research was funded by the University of Johannesburg’s University Research Committee (URC) grant for Kingsley A. Ogudo (2019), and the Department of Electrical and Electronic Engineering Technology (Kingsley A. Ogudo) Research cost center and the APC was funded by a grant from the University of Johannesburg Library Research Funds (UJ). The funders had no role in study design, data collection and analysis, decision to publish, or preparation of the manuscript.

==============================
Antigenic peptides (APs), also known as T-cell epitopes (TCEs), represent the immunogenic segment of pathogens capable of inducing an immune response, making them potential candidates for epitope-based vaccine (EBV) design. Traditional wet lab methods for identifying TCEs are expensive, challenging, and time-consuming. Alternatively, computational approaches employing machine learning (ML) techniques offer a faster and more cost-effective solution. In this study, we present a robust XGBoost ML model for predicting TCEs of the severe acute respiratory syndrome coronavirus 2 (SARS-CoV-2) virus as potential vaccine candidates. The peptide sequences comprising TCEs and non-TCEs retrieved from Immune Epitope Database Repository (IEDB) were subjected to feature extraction process to extract their physicochemical properties for model training. Upon evaluation using a test dataset, the model achieved an impressive accuracy of 97.6%, outperforming other ML classifiers. Employing a five-fold cross-validation a mean accuracy of 97.58% was recorded, indicating consistent and linear performance across all iterations. While the predicted epitopes show promise as vaccine candidates for SARS-CoV-2, further scientific examination through in vivo and in vitro studies is essential to validate their suitability.

Introduction

The outbreak of COVID-19 caused by the severe acute respiratory syndrome coronavirus 2 (SARS-CoV-2) led to a global health crisis (Chakraborty et al., 2020). Since its emergence in late 2019, the virus has spread rapidly, resulting in large number of confirmed cases and deaths worldwide. As of the latest available data, there have been millions of reported cases and a significant number of deaths attributed to COVID-19 around the world (Worldometer, 2020). The WHO has been at the forefront of monitoring and responding to the COVID-19 pandemic and has provided guidelines and recommendations for governments, healthcare systems, and individuals to prevent the spread of the virus, mitigate its impact, and develop effective interventions (WHO, 2022). Understanding the molecular biology of SARS-CoV-2, including its variants, and the interaction with the immune system is crucial in devising effective strategies against the pathogen. In terms of taxonomy, Coronaviruses (CoVs) are classified as members of the Orthocoronavirinae subfamily within the Coronaviridae family, which is part of the Cornidovirineae sub-order of the Nidovirales order (Pal et al., 2020). CoVs are enveloped RNA viruses with a genome size of ~30 kb, and they are known for causing respiratory and gastrointestinal infections in humans and animals (Pal et al., 2020).

The genetic code of the virus contains various proteins, both structural and non-surface ones. The former type includes envelope-E, surface-S, nucleocapsid-N, and membrane-M proteins, while the latter type comprises orf1ab polyproteins (Su et al., 2016). Due to their sequence similarity, non-surface proteins are considered potential targets for the activation of cytotoxic CD8+ T-cells, which can help protect against the virus (Khailany, Safdar & Ozaslan, 2020). Both the innate and adaptive immune systems participate in defending against SARS-CoV-2 infection. CD4+ and cytotoxic CD8+ T cells play crucial roles in eradicating the virus (Liu et al., 2019). SARS-CoV-2 enters cells through the binding of its spike receptor-binding domain (RBD) to the human angiotensin-converting enzyme 2 (ACE2) receptor in concert with accessory receptors/molecules that facilitate viral attachment, internalization, and fusion (Zhang et al., 2020). While angiotensin-converting enzyme 2 (ACE2) is the established primary receptor for SARS-CoV-2 attachment and entry into host cells, studies have identified alternative receptors, that could contribute to viral infection in specific tissues or under certain conditions, even if ACE2 expression is low. The other receptors and co-receptors include NRP-1, CD147, CD209L, asialoglycoprotein receptor 1 (ASGR1), KREMEN1 (kringle containing transmembrane protein 1) protein, heparan sulfate proteoglycans (HSPG) syndecan and glypican, and receptor tyrosine kinase AXL, high-density lipoprotein type 1 scavenger receptor (SR-B1), GRP78 (glucose-regulated protein 78) (Avdonin et al., 2023). Several variants of SARS-CoV-2 have emerged over time with different genetic mutations, some of which are associated with increased transmissibility and potential for immune evasion (Niessl, Sekine & Buggert, 2021). The Omicron variant (B.1.1.529) of SARS-CoV-2 has caused alarm due to its potential to spread more easily and affect the efficacy of vaccines (Harvey et al., 2021). Understanding the molecular characteristics of this variant and its implications for vaccine development is crucial in addressing the ongoing challenges posed by the pandemic. The spike protein is the primary means through which the virus interacts with the immune system, and it is the main target of COVID-19 vaccines (Lee et al., 2021).

In light of the continuous efforts to contain SARS-CoV-2 infections and manage virus mutations, it is vital to employ epitope-based vaccine (EBV) development approaches (Huber et al., 2014). EBVs involve the use of antigenic peptides (APs), known as T-cell epitopes (TCEs), which are the specific regions of a pathogen’s proteins that trigger an immune response (Seder, Darrah & Roederer, 2008). These TCEs are recognized by human leukocyte antigen (HLA) molecules on infected cell surfaces and are important for characterizing T-cell immunity and developing vaccines and immuno-therapies (Liu et al., 2019). Compared to traditional vaccine approaches, EBVs have several advantages, including improved safety, immunogenicity, faster production times, and cost-effectiveness. Identifying suitable TCEs as vaccine candidates is the main step in the development of EBVs (Grifoni et al., 2020). TCEs can be identified using traditional wet lab approaches, which involve in vitro assays and animal studies. However, these methods can be labour-intensive, time-consuming, and costly (Lee et al., 2021). In recent years, computational techniques, particularly ML, have emerged as a promising alternative for epitope prediction due to their ability to rapidly process large amounts of data and identify patterns in peptide sequences (Bukhari et al., 2022). In this study, we propose an ML-based computational model using XGBoost classifier, a powerful variant of the boosting technique known for its computational efficiency and scalability. The advantage of using XGBoost is that it can be trained relatively quickly on large datasets without requiring extensive computational resources, unlike many deep learning (DL) models that may need substantial computational power and longer training times. In addition, DL models typically require large amounts of data to achieve optimal performance. In contrast, XGBoost can perform well even with moderate-sized datasets, which is often the case in biological data where obtaining large labelled datasets can be challenging. Furthermore, XGBoost allows for effective feature engineering and has robust mechanisms for handling overfitting through regularization parameters and cross-validation. Although ML models also employ regularization techniques, they are more prone to overfitting, especially with smaller datasets. The predicted APs/TCEs using the proposed model may possess a high probability to act as vaccine targets subjected to in vivo and in vitro tests. The proposed model developed would prove helpful to scientific fraternity working in epitope-based vaccine designing save time to screen the active TCE candidates of SARS-CoV-2 pathogen against the inactive ones.

Motivation

Using whole-organism vaccines poses several challenges, especially for immunocompromised individuals. EBVs can be utilized to overcome the issues associated with multicomponent and heterogeneous vaccines. They serve as effective alternatives due to their cost-effectiveness and reduced potential for side effects like allergies. In addition, most of the current methods like NetMHC (Nielsen et al., 2003) focus on predicting peptide-binding probability, which fail to identify whether a peptide sequence is TCE or non-TCE in a deterministic approach. CTLpred (Bhasin & Raghava, 2004), for instance, predicts epitopes in a deterministic manner but is limited to peptides having length upto 9-mers. Therefore, a direct approach to TCE prediction has been proposed here, which resolves the first problem. The proposed model is also capable of predicting epitopes of varying lengths (beyond 9-mers), fixing the second problem associated with the existing methods. The focus of the study is on predicting TCEs that can activate cytotoxic T lymphocytes (CTLs) because of the two main reasons: 1. CTLs directly target and eliminate virus-infected cells, offering a strong defense mechanism against viral replication. 2. Also, CTL activation can lead to the development of memory T cells, contributing to long-term immune protection against SARS-CoV-2.

Contributions

This research introduces several novel contributions. Primarily, the study proposes a robust XGBoost ML model for predicting TCEs of the SARS-CoV-2 virus that can activate cytotoxic T lymphocytes (CTLs). This emphasis stems from the well-established role of CTLs in mediating cellular immunity and directly eliminating virus-infected cells. The boosting approach used in the current study not only enhances prediction accuracy but also provides deeper insights into the physicochemical properties of antigenic peptides, which are crucial for EBV design. By systematically evaluating a comprehensive dataset of SARS-CoV-2 peptide sequences, we have demonstrated the efficacy of our model in identifying potential TCEs with high precision and recall rates. Secondly, the research prioritized achieving high accuracy in TCE prediction, with the proposed model demonstrating impressive accuracy. Moreover, K-fold cross-validation (KFCV) was performed, confirming the model’s reliability and consistency for the prediction of TCEs across all folds. The predicted epitopes could serve as the key targets for developing a PBV.

The manuscript is structured as follows: the related work section offers an extensive review of literature, the methodology section explains the proposed methods, the evaluation metrics used to evaluate the model are explained in model evaluation section, the results are presented and discussed in results section and the conclusion section wraps up the manuscript.

Literature review

Following the release of the SARS-CoV-2 genome sequences publically in early 2020, researchers swiftly turned to ML techniques with the aim of discerning potential vaccine candidates against the pathogen, focusing particularly on analyzing TCEs with antigenic properties (Sohail et al., 2021). A study by Kim et al. (2024) investigates the diversity of CD8+ T-cell epitopes essential for long-term immunity against SARS-CoV-2 by analyzing 27 common HLA-A and HLA-B alleles across 16 virus variants. The research found high conservation of epitopes in spike, membrane, and nucleocapsid proteins, though newer variants like Omicron BQ.1-XBB.1.5 showed reduced binding affinity in spike epitopes. Additionally, the study identified specific HLA alleles that might influence susceptibility or protection against severe COVID-19, emphasizing the role of T-cell responses in developing long-term immunity. Bravi (2024) explores the role of ML in rational vaccine design, specifically in identifying B and T cell epitopes and correlates of protection. The author argues that interpretable ML can enhance immunogen identification and scientific discovery by revealing molecular processes in vaccine-induced immune responses. Alibakhshi et al. (2024) in their study addressed the need for a more robust SARS-CoV-2 vaccine by developing a multi-epitope protein vaccine using four key viral proteins. Researchers selected antigenic determinants from these proteins and designed a protein structure with four domains containing epitopes. The designed structure was then evaluated for its antigenic potential, three-dimensional structure, and interaction with immune system receptors through molecular docking and dynamic molecular (MD) simulation. The results demonstrated good relative stability and effective interactions with receptors, suggesting the designed structure is a promising vaccine candidate for further research. Federico et al. (2023) employed an AI-driven tool, the NEC Immune Profiler (NIP), to predict T cell immunogenicity hotspots across the SARS-CoV-2 viral proteome for developing universal T cell vaccines. Using a flow cytometry-based T cell activation-induced marker (AIM) assay, researchers validated 59 epitopes, 56% of which were novel and primarily derived from non-spike regions (Orf1ab, Orf3a, and E). Additionally, spike protein peptides predicted by NIP induced CD8+ T cell responses in vaccinated donors’ PBMCs. The findings support the predictive accuracy of AI models and offer a new framework for assessing vaccine-induced T cell responses. A study (Erez et al., 2023) highlights the importance of T-cells in immune responses to SARS-CoV-2 and the need for reliable assays to evaluate cellular immunity post-infection or vaccination. Researchers developed a cellular activity assay based on short peptide presentation to identify T cell epitopes on different MHC alleles. Using peptide libraries and computational scanning, they identified four CD8 T cell epitopes, including one novel epitope, from the SARS-CoV-2 spike protein. The assay was validated in various mouse models, demonstrating its potential to inform antigen design for vaccines and establish assays for emerging pathogens. Crooke et al. (2020) devised a computational approach to identify the SARS-CoV-2 proteome, utilizing open-source web tools and algorithms to identify potential B and T-cell epitopes. Their method led to the identification of 41 B-cell and 6 T-cell epitopes considered promising targets for EBV design against the SARS-CoV-2 virus. Dong et al. (2020) focused on the creation of a multi-epitope vaccine for preventing and treating COVID-19, employing immunoinformatics techniques. They synthesized epitopes from B cells, cytotoxic T lymphocytes (CTLs), and helper T lymphocytes (HTLs) of SARS-CoV-2 proteins, incorporating linkers to construct the vaccine. Additionally, they enhanced the vaccine’s immunogenicity by attaching a 45-mer peptide sequence known as β-defensin and a pan-HLA binding peptide (13aa) to the N-terminus of the vaccine using an EAAAK linker. Bukhari et al. (2021) proposed a novel ensemble ML technique for identifying TCEs of the SARS-CoV-2 virus, achieving a notable accuracy of 98.20% and an AUC of 0.991. Mahajan et al. (2021) developed a novel TCR-binding algorithm to identify CD8 TCEs present in the spike antigen of SARS-CoV-2. By employing this approach, they identified predicted epitopes that elicited robust T-cell activation, revealing the presence of pre-existing CD4 and CD8 T-cell immunity against the virus’s antigen. Rencilin et al. (2021) employed a method to narrow down the pool of potential CTL epitopes to 50 peptide sequences, 45 of which were common among all the SARS-CoV-2 strains they tested, thereby enhancing the likelihood of successful vaccine development. Meyers et al. (2021) identified TCEs from the envelope, membrane, and spike segments of the virus, observing that 97% of the analyzed epitopes elicited a robust T-cell response. Furthermore, Fatoba et al. (2021) utilized established epitope prediction methods to identify 19 CD8 and 18 CD4 epitopes with the potential for developing an effective vaccine against SARS-CoV-2. Cihan & Ozger (2022) proposed a hybrid model that integrates various ML classifiers to identify non-allergenic and non-toxic epitopes. These epitopes hold promise for the development of COVID-19 vaccines and protection against other SARS-CoV viruses in the future, with the model achieving promising results featuring an AUC of 94.0%. The T-cell and B-cell epitopes identified for SARS-CoV-2 through bioinformatics tools, immunoinformatics approaches, and ML models offer valuable insights for developing epitope-based vaccines. These identified epitopes possess the potential to evoke a robust immune response and may play a pivotal role in the development of effective EBVs against the SARS-CoV-2 virus. A study by Yang, Bogdan & Nazarian (2021) introduces an in-silico deep learning method called DeepVacPred, designed to create a multi-epitope vaccine targeting both B-cell and T-cell immunity. DeepVacPred analyzes the SARS-CoV-2 spike protein sequence and predicts 26 potential vaccine subunits. These subunits are then further analyzed for B-cell, CTL, and HTL epitopes, leading to the selection of the best 11 for the vaccine. The resulting 694-amino acid multi-epitope vaccine is predicted to contain B-cell, CTL, and HTL epitopes, potentially offering comprehensive protection against SARS-CoV-2 infection.

Methodology

The methodology for constructing the proposed XGBoost model to predict TCEs of the SARS-CoV-2 virus is presented in Fig. 1 and is described in detail through the following subsections.

Figure 1 The proposed methodology workflow.

Created using https://app.diagrams.net.

Sequences retrieval

Choosing experimentally confirmed antigenic TCEs is crucial for developing an EBV against SARS-CoV-2. For this research, the peptide sequences comprising TCEs and non-TCEs were obtained from the Immune Epitope Database (IEDB) repository in CSV format (Vita et al., 2019). All the peptides are linear and 17,951 sequences were collected, comprising 8,883 TCEs and 9,068 non-TCEs. The length of peptides ranges from 7 to 45 mers (no. of amino acids). To solve the binary classification issue, the “Class” target variable was added to both CSV files. A value of one was assigned to epitope sequences, while non-epitope sequences were given a value of 0.

Data cleansing and feature extraction

Amino acids are the building blocks of proteins and peptides, and each amino acid possesses unique physicochemical properties that influence its biological functions (Cruse et al., 2004). These properties include polarity, hydrophobicity, charge, size, shape etc. In determining the structure and function of proteins, the physicochemical properties of amino acids play a pivotal role.

Understanding these properties is essential for predicting protein-protein interactions, designing peptide-based drugs, and developing new protein-based materials. Before performing Feature Extraction (FE), the duplicate entries and nonlinear sequences were removed. For each peptide sequence, these physicochemical properties are considered as features or independent variables in the current study. To extract features from peptide sequences, we utilized the peptides (Osorio, Rondón-Villarreal & Torres, 2015) and peptider (peptider, 2015) packages available in R programming environment (R Core Team, 2013). These packages include numerous functions for calculating different indices and physicochemical properties of amino acid sequences. The following features or physicochemical properties (PPs) were utilized in this study. Aliphatic Index (AI)

Boman Index (BI)

Insta Index (II)

Probability of Detection (PD)

Hmoment Index (HMI)

Molecular Weight (MW)

Peptide Charge (PC)

Hydrophobicity (H)

Isoelectric Point (IP)

Kidera Factors (KF)

Amino Acid Composition (AAC)

A summary of the physicochemical properties and their notations utilized in the current study are provided in Table 1. The PPs namely PC, H, IP and KF were measured on 20, 20, 9 and 10 scales respectively. The functions corresponding to each physicochemical property inside peptides and peptider packages allowed us to efficiently extract essential information about the amino acid sequences, which is critical for the development of peptide-based vaccines and drugs. The FE process resulted in a high-dimensional dataset of 68 features for each sequence in the CSV files.

Table 1 Physicochemical properties.

Property	Notation used	Count	
AI	F1	1	
BI	F2	1	
II	F3	1	
PD	F4	1	
HMI	F5_1, F5_2	2	
MW	F6_1, F6_2	2	
PC	F7_1 to F7_20	20	
H	F8_1 to F8_20	20	
IP	F9_1 to F9_9	9	
KF	F10_1 to F10_10	10	
AAC	F11	1	

Feature selection

Feature selection (FS) is a technique that involves selecting a subset of relevant features from a larger set of features extracted from a dataset. The significance of FS is its capability to enhance the precision and efficiency of machine learning models by eliminating irrelevant features. By selecting only the most informative features, we can eliminate noise and improve the performance of our ML models (Bukhari, Webber & Mehbodniya, 2022). Additionally, FS also help in reducing overfitting, the computational cost and memory requirements of the machine learning algorithms, making them more practical for real-world applications. For the current study, the Boruta algorithm (Kursa & Rudnicki, 2010) in the R programming environment was used to identify the most relevant and important features from a given dataset by comparing the feature importance of the original features with that of their shadow attributes. The shadow attributes are essentially random permutations of the original features used to create a null distribution of feature importance. The Boruta algorithm works by iteratively comparing the importance of each feature to that of its shadow attributes using a random forest model. The importance of a feature is calculated based on the decrease in mean accuracy or increase in mean squared error when the feature is excluded from the model. The shadow attributes that have a higher importance than the original feature are marked as “uninformative,” indicating that the original feature is redundant or irrelevant. The algorithm continues to iterate until all features have been marked as informative or uninformative (Kursa & Rudnicki, 2010). The final set of informative features is then selected based on their rankings of importance, with the most important features selected first. The Boruta algorithm takes the outcome variable “Class” and 78 features as input for feature selection. The algorithm returned the top 15 features based on their Mean Decrease Accuracy (MDA) score. Equation (1) presents the target variable “Class” and the 20 most relevant features that were used for training the model. The selected important features as shown in Eq. (1) are arranged in decreasing order of importance score. These features were selected based on their high importance scores and are expected to have a significant impact on the prediction performance of the model.

(1) Class∼f(F3,F5,F2,F6_2,F8_4,F7_1,F1,F9_3,F7_6,F11,F10_3,F8_2,F8_14,F7_11,F10_1)

Model building

The proposed model is built using the Extreme Gradient Boosting (XGBoost) classifier commonly used in predictive modeling. XGBoost, an enhanced version of gradient boosting, not only provides superior computational speed but also enhances scalability through parallel learning with multiple CPU cores. XGBoost-an ensemble learning method that combines multiple weak prediction models for robust predictions (Budholiya, Shrivastava & Sharma, 2022), employs an approach similar to decision tree building process. This process involves partitioning the data into subsets based on specific feature values, as depicted in Fig. 2.

Figure 2 Working of XGBoost.

The working of XGBoost is explained through the following steps:

Step 1: To begin the modeling process, XGBoost starts with a single decision tree, known as the initial or base model.

Step 2: To assess the variance between the predicted and actual values, a loss function is employed. XGBoost employs a gradient boosting technique, which involves minimizing the gradient of the loss function in relation to the model parameters (Torlay et al., 2017).

Step 3: To expand the model, XGBoost gradually introduces additional trees, one by one, by training each new tree to rectify the errors of the preceding ones. The algorithm adopts a greedy methodology to divide each tree node, selecting the feature that results in most significant reduction in the loss function.

Step 4: To avoid over-fitting, XGBoost utilizes various methods, including limiting the depth of each tree, introducing penalties to the loss function to discourage excessive model parameters, and sub-sampling both training data and features.

Step 5: After training all the trees, XGBoost merges their predictions to generate the outcome. The technique employs a weighted combination of the separate tree results, where the weights are established by the tree depth and learning rate.

Step 6: The XGBoost algorithm keeps adding additional trees until it fulfils a predefined stopping criterion, like attaining a maximum limit of trees or accomplishing a minimum advancement in the loss function.

Mathematical modelling

Mathematically, a general gradient boosting (GB) involves fitting the residual by utilizing the negative gradient of the model with respect to the data as an estimate of the residual. On the other hand, XGBoost also fits to the residual of the data but it utilizes the second-order Taylor expansion to fit the loss residuals of the model. It further enhances the loss function of the model by incorporating a regularization term for the complexity of the model. Let D represent a dataset as defined in Eq. (2).

(2) D={(xi,yi|xi∈Rm,yi∈R,i=1,….,n}

where xiis a set of m features, yiis a real value, and i represents the index of n total samples, we can predict a cumulative value (y^i) by using k decision trees that rely on the input xi. Mathematically this prediction is represented by Eq. (3) where m is the number of features in the data set, n is the number of samples and F is the set of trees.

(3) y^i=φ(xi)=∑k=1kfk(xi),fk∈F.

XGBoost has an objective function that incorporates both a loss function and a complexity term, which is aimed at avoiding overfitting. The complexity term is composed of two components, namely the number of leaves and the L2 regularity.

(4) L(Ø)=∑il(y^i,yi)+∑kΩ(fk)

where the term Ω(f)=γT+12λ||ω||2 is capable of producing a minimum value due to the convex nature of both the loss and complexity terms. To put it simply, ω represents the difference between the actual and predicted values, and by adding the L2 regularity of ω to the objective function, overfitting can be avoided.

Once the objective function is established, the focus shifts to the training process. In each iteration, the training of the tree’s objective function can be expressed in the following manner:

(5) L(t)=∑i⁡l(yi,y^it−1)+ft(xi)+Ω(ft).

The predicted value from the previous round (t − 1) is taken as input, and the residual is fitted using the actual value. The formula involves using a second-order Taylor expansion approximation to derive the form of the loss function.

(6) L(t)≈∑i⁡l(yi,y^it−1)+gift(xi)+12hift2(xi)+Ω(ft)

where gt=∂y^(t−1)l(yi,y^(t−1)),hi=∂y^(t−1)2l(yi,y^(t−1)).

Following a series of computations, the minimum objective function value for ω is expressed as follows:

(7) ωjt=−∑i∈Ijgi∑i∈Ijhi+λ.

Once substituted back into the original equation, the minimum value that is attained is:

(8) L∼(t)(q)=−12∑j=12(∑i∈Ijgi)∑i∈Ijhi+λ+λT.

Therefore, the ω that is ultimately derived represents the optimal solution for the objective function under the given set of sample conditions.

Hyperparameter tuning

Hyperparameter tuning (HPT) involves choosing the most effective values for hyperparameters that impact a machine learning model’s performance. These hyperparameters are established before the commencement of model training and are not acquired from the data. The objective of HPT is to identify the optimal combination of hyperparameters that minimizes the model’s loss function on the validation dataset. This entails iteratively adjusting the hyperparameter values and training the model until the optimal set of hyperparameters is determined. Following are the hyperparameters of the XGBoost algorithm that were tuned to improve the model’s performance (XGBoost, 2022). Learning rate (LR): The LR plays a critical role in determining the size of each step taken during every iteration of the gradient boosting process. We experimented with various LRs ranging from 0.01 to 0.1 and eventually opted for a value of 0.05.

Maximum depth (MD): The MD of each tree controls the complexity of the model and the degree to which it can capture interactions between features. We tested several values for the MD, ranging from 3 to 7, and ultimately selected a value of 5.

Number of trees (NoT): The NoT in the ensemble determines the overall complexity of the model and its ability to capture complex relationships between features. We tested several values for the NoTs, ranging from 50 to 300, and a value of 200 was chosen to be the optimal value.

Subsample ratio (SR): The SR parameter controls the fraction of the training data that is used to train each tree in the ensemble. We tested several values for the SR, ranging from 0.5 to 0.9, and a value of 0.8 was used finally.

Minimum child weight (MCW): The MCW parameter is responsible for managing the minimum total of instance weight required in a child. It is also used to prevent overfitting. We tested several values for the MCW, ranging from 1 to 20, and ultimately selected a value of 10 based on the model performance on the validation data.

For hyperparameter tuning, we employed the grid search technique. Grid search hyperparameter tuning entails constructing a grid of every possible combination of hyperparameters and conducting a comprehensive search of the grid to determine the combination of hyperparameters that yields the optimal model performance (XGBoost, 2022). The results of the grid search hyperparameter tuning technique are presented as Appendix 1. Algorithm S1 outlines the hyperparameter tuning process using a grid search conducted in this study for the proposed XGBoost classifier.

Note that the cartesian_product function is used to generate all possible combinations of hyperparameters. This function takes in an arbitrary number of alterable arguments and returns the Cartesian product of those iterables as a generator. The combination of hyperparameters that produces the best performance is then selected as the optimal combination.

Models used for comparative analysis

We conducted a performance comparison of the proposed model with other state-of-the-art existing classifiers, namely support vector machine (SVM), decision tree (DT), neural network (NN), random forest and logistic regression (LR). The models were implemented using the R programming language in accordance with the guidelines of the “GNU General Public License” (R Core Team, 2013). The following is a brief description of the models used for the comparative analysis in this study.

Support vector machine

SVM works by finding a hyperplane (a separating line) that best separates data points into their respective classes. The best hyperplane is the one that maximizes margin i.e., the distance between itself and the observations closest to it, also called as support vectors (Shmilovici, 2005). Let X represent a set of input variables i.e., physicochemical properties, y be the corresponding target variable i.e., epitope or non-epitope, which can take two values: 1 and 0 respectively, and v is a vector perpendicular to the hyperplane whose magnitude represents the margin. The mathematical representation of hyperplane is given in Eq. (9).

(9) H:vX+c=0

where c is the distance of v from origin to the hyperplane. If all data points in X are at distance larger than one from the hyperplane, these two rules hold for every sample {xi, yi} in the dataset:

(10) vxi+c≥1ifyi=1

(11) vxi+c≤−1yi=−1.

Since the goal of SVM is to maximize the margin for a particular hyperplane, it falls into the category of quadratic optimization problem.

(12) Maximize:1|v|

such that for every point xi in X to be correctly classified the below condition should always hold.

(13) yi(v⋅xi+c)≥1∀(xi,yi).

Decision tree

DT works by recursively splitting the given dataset into a tree-like structure based on the most informative features. At each node (partition), it chooses the best feature to divide the data and eventually terminates at leaf nodes, which provide the target values. The internal nodes represent the features of the dataset, branches between the nodes represent the prediction rules (or decision rules) and finally the leaf nodes represent the target variables (output) (Song & Lu, 2015).

Let X be the root node of DT that represents the whole dataset and F be the set of features present in X. To split the data, a feature and a threshold must be chosen that best separates the data into two or more subsets. To quantify the ability of each attribute in F to discriminate between classes multiple attribute selection measures (ASM) can be used including Gini impurity (GI), entropy (E) or information gain (IG). GI is a measure of the disorder or impurity in a set of data. For a classification task with multiple classes, the Gini impurity is defined using Eq. (14).

(14) GI(X)=1−∑n=1tpn2

where t is the number of classes, pn is the proportion of data points in class n in the dataset X.

While using GI as ASM, select the feature that minimizes the GI when used for splitting.

E is another measure of disorder or impurity. Equation (15) defines entropy for a classification task with multiple classes.

(15) E(X)=−∑n=1tpn2.log2pn.

IG is used to determine the effectiveness of a feature in reducing uncertainty about the classification. It is calculated as the difference between the entropy (or GI) before and after a split on a feature using Eq. (16).

(16) IG(X,F)=E(X)−∑u∈Values(Fi)XuX.E(Xu)

where Fi is the feature being evaluated and Values (Fi) represents the values of that feature. Xu is the subset of X for which feature Fi has value u. While using IG as ASM select the attribute that maximizes IG.

Neural network

NN is a supervised ML classifier that comprises interconnected layers of nodes called as neurons, which process input data to produce an output. Within each neuron, a mathematical function is applied to the weighted sum of its inputs, and the resulting output is then transmitted to the next layer. This process continues until the final output is produced (Han et al., 2018). These neural networks undergo training using an optimization algorithm designed to minimize a loss function. This loss function quantifies the disparity between the predicted output and the actual output during the training process. Let x be the input vector such that

x=[x1,x2,…,xn]t.

If y is the output of a neuron in a NN then

(17) y=f(c+wx)

where σ is the activation function applied to the weighted sum of inputs, w is the weight associated with each input term x, and c is the bias term.

Random forest

RF is a modification of bagging that combines a large number of de-correlated DTs so that each tree captures different aspects of data and variance of resulting model is reduced. It then averages their predictions to produce the final output (Breiman, 2001). For most of the tasks the performance of RFs is analogous to boosting but they are comparatively simpler to train and fine-tune. Prediction using RF involves the following steps: Data sampling (bootstrapping): Randomly select a subset d from the dataset D and create a DT using the bootstrap sample d.

Feature selection: Before each node split in an individual DT, a random subset p of input features F is considered as candidates for splitting. This introduces variability in feature selection. Usually the value of p is set to F or values as low as 1.

Bagging: Build N such DTs using N samples from the dataset and combine them as {DTn}1N

Prediction: Since TCE prediction is a classification problem, the final output is decided by majority voting where the class with most votes from DTs in RF is selected as the final prediction of RF.

(18) y(xi)=argmaxc∑n=1NI(DT(x;Θb)=c)

where argmaxc finds the class with the highest cumulative count among all the predictions and I() is the indicator function that evaluates to 1 when the condition is true and 0 otherwise.

Logistic regression

LR is a statistical method used for binary classification problems, where the goal is to predict one of two possible outcomes. Unlike linear regression, which predicts a continuous output, LR predicts the probability that a given input belongs to a particular category (Bisong, 2019). The core of LR is the logistic function, also known as the sigmoid function as shown in Eq. (19), which maps any real-valued number into a value between 0 and 1. This is crucial for predicting probabilities.

(19) σ(z)=11+e−z

where “z” is the linear combination of input features (also known as the logit).

The input features (independent variables) are combined linearly using weights (coefficients) and a bias term. This combination is then passed through the sigmoid function as shown in Eq. (20).

(20) z=w0+w1x1+w2x2+…+wnxn

where w0 is the bias term, wi are the weights, and xi are the input features.

The sigmoid function output represents the probability as shown in Eq. (21) that the given input belongs to a particular class (e.g., TCE (1) or non-TCE (0)).

(21) y^=σ(z)=11+e−z.

To make a binary decision (0 or 1), a threshold is applied to the probability output. Typically, a threshold of 0.5 is used, meaning if the predicted probability is greater than or equal to 0.5, the output is class 1; otherwise, it is class 0.

Model evaluation

Model evaluation (ME) refers to the process of assessing and measuring the performance of a trained ML model on new, unseen data (Fatourechi et al., 2008). The various metrics (Suryanarayanan et al., 2019) used in this study to evaluate the proposed model are:

(22) Accuracy=(TP+TN)/(TP+TN+FP+FN)

(23) Recall=TP/(TP+FN)

(24) Precision=TP/(TP+FP)

(25) F1score=2∗(precision∗recall)/(precision+recall)

AUC: The “Area under the Curve” (AUC) plots the true positive rate (TPR) against the false positive rate (FPR) for different classification thresholds. The AUC ranges from 0 to 1, where 0 represents a model that always predicts the wrong class, and 1 represents a model that always predicts the correct class.

Where “TP: True Positive, TN: True Negative, FP: False Positive and FN: False Negative”

Models with a large number of parameters relative to the amount of training data are more prone to overfitting. This complexity allows the model to fit noise in the training data, leading to inflated performance metrics. When the dataset is small relative to the complexity of the problem, the model may memorize the training examples rather than learning underlying patterns. This memorization results in poor generalization to new data. To mitigate overfitting several steps can be taken like regularization, use of efficient FS techniques, cross-validation etc. Regularization techniques like L1 or L2 regularization penalize large weights in the model, encouraging it to be simpler and less prone to fitting noise. This helps in improving generalization. Choosing relevant features and reducing irrelevant or noisy ones can simplify the model and improve its ability to generalize. Implementing these strategies can effectively address overfitting concerns and improve the robustness of their models to new data. In this study apart from employing a robust FS process, a technique known as K-fold cross-validation (KFCV). It entails partitioning the dataset into ‘k’ folds or subsets of roughly equal size. Subsequently, the model undergoes training ‘k’ times, with each iteration utilizing a different fold as the testing set and the remaining folds as the training set. This iterative process continues until each fold serves as the testing set exactly once, as illustrated in Fig. 3. The accuracy attained in each run, denoted as Acc(i), is then averaged, and the model’s average accuracy is computed using Eq. (26).

(26) AverageAccuracy=1k∑i=1kAcc(i).

Figure 3 K-fold cross validation process.

Results

In this section, we will discuss the results obtained by the proposed XGBoost model and the existing classifiers. The test set which is 20% of the total dataset was used to evaluate the performance of the proposed model and other classifiers. The results obtained by proposed model and the existing classifiers are presented in Table 2.

Table 2 Results obtained by the proposed model and other classifiers.

Model	Accuracy (%)	Precision	Recall	F-score	AUC	
Decision tree	95.66	0.941	0.951	0.932	0.976	
Neural network	94.19	0.935	0.958	0.931	0.949	
SVM	95.32	0.949	0.952	0.937	0.979	
Random forest	96.98	0.942	0.961	0.946	0.981	
Logistic regression	92.41	0.911	0.928	0.903	0.936	
Proposed model	97.6	0.963	0.971	0.958	0.99	

From the results, it is clear that the XGBoost model achieved an accuracy of 97.6% which indicates that the model is highly accurate in predicting TCEs of SARS-CoV-2. A precision of 0.963 indicates that the model has a low false positive rate, meaning that it does not predict epitopes that are not actually TCEs of SARS-CoV-2, and a recall of 0.971 indicates that the model has a low false negative rate, meaning that it does not miss many actual T-cell epitopes of SARS-CoV-2. In addition, a value of 0.958 for the F1 score suggests that the model has a good balance between these two metrics. The AUC curve is depicted in Fig. 4 and a value of 0.99 for the AUC indicates that the model has high discriminatory power and is able to accurately distinguish between TCEs of SARS-CoV-2 and non-TECs.

Figure 4 AUC curve of the proposed model.

Examining the model’s dependability and consistency is crucial; is it prone to underfitting or overfitting issues? To assess this, a five-fold CV (5FCV) was performed (K = 5 in the current study) (Bukhari, Jain & Haq, 2022). The iteration-specific accuracies obtained have been illustrated in Table 3. The average accuracy attained through 5FCV, which is 97.58%, is promising, and it is evident that the XGBoost model performs consistently well across all folds. Based on the evaluation results (presented in Table 2) using the test dataset and conducting the KFCV technique (results presented in Table 3) the findings suggest that the XGBoost model proposed in this study outperforms the existing classification models. This in turn suggests that the proposed XGBoost based ML model is reliable in predicting TCEs of SARS-CoV-2, with high accuracy, precision, recall, F score, and AUC.

Table 3 Accuracies obtained using K-fold cross validation.

Runs	DT	NN	SVM	RF	XGBoost	
1	95.98	93.41	93.76	94.19	97.65	
2	96.01	93.62	94.77	95.76	98.23	
3	95.34	94.01	95.19	96.21	98.01	
4	95.99	93.17	94.83	93.89	97.12	
5	95.59	94.53	94.62	96.78	96.89	
Average accuracy	95.78	93.74	94.63	95.36	97.58	

To summarize, the proposed model utilizes the XGBoost algorithm, which has been demonstrated to outperform traditional classification algorithms such as SVM, DT, RF, and NN in terms of prediction accuracy and precision. Specifically, the proposed model achieved an accuracy of 97.6% and a precision of 0.963, surpassing the performance metrics reported in recent studies. XGBoost is known for its computational efficiency due to its implementation of parallel processing and tree pruning. This makes it faster and more scalable than many ML models, which require extensive computational resources and longer training times. The application of feature selection with an aim to significantly reduce the dimensionality of the dataset not only speeds up the training process but also minimizes the risk of overfitting, contributing to a more generalizable model.

Comparative analysis with benchmark techniques

To contextualize the performance of the proposed model, we have compared it with two benchmark techniques: NetMHC and CTLpred. NetMHC is designed to estimate peptide-binding capacity to MHC molecules. While it effectively predicts the binding potential, it does not directly predict whether a particular peptide is a SARS-CoV-2 TCE or non-TCE. This indirect approach can lead to less accurate identification of true epitopes. The proposed model directly predicts TCEs, specifically addressing this limitation. This direct approach improves the accuracy and relevance of epitope predictions for SARS-CoV-2.CTLpred directly predicts cytotoxic T lymphocyte (CTL) epitopes. The major constraint with CTLpred is its limitation to peptide sequences of nine (9) amino acids in length (9-mers). This restriction can overlook longer epitopes that are crucial for immune response. The proposed model overcomes this limitation by predicting epitopes of variable lengths (7 to 15 mers), including those greater than 9-mers. This flexibility allows for a more comprehensive identification of potential epitopes. Since the NetMHC server only estimates the binding capacity of a peptide sequence, as shown in the third column of Table 4, the proposed model offers greater efficiency by deterministically predicting whether a peptide is an epitope or not. The proposed model provides discrete predictions, indicating either 1 (TCE) or 0 (non-TCE), as seen in the last column of Table 4. While the CTLpred server also offers discrete predictions, it is limited to sequences up to 9-mers in length. As illustrated in column 4 of Table 4, CTLpred cannot predict sequences longer than 9-mers, represented by hyphens (-). In contrast, the proposed model accurately classifies sequences longer than 9-mers as SARS-CoV-2 TCE or non-TCE. The results depicted in Table 4 indicate that the proposed model achieves exceptional accuracy (100% in this case) by correctly classifying all peptide sequences. This comparison demonstrates that the proposed model outperforms existing techniques.

Table 4 Comparative analysis of the proposed model with the existing benchmark methods.

Peptide sequence	Actual class
(1-TCE and 0-non-TCE)	Prediction by NetMHC	Prediction by CTLpred
(1-TCE and 0-non-TCE)	Prediction by the proposed model	
ATCAITHK	1	42	1	1	
QRPADCAT	1	64	1	1	
MQNLTLTEIRVSQDP	1	7.2	–	1	
VTRHSPRDPSKPQYN	1	4.1	–	1	
SQPSRSM	1	74	1	1	
LSKTVEHPR	0	4.1	0	0	
SVKHVYQL	0	47	0	0	
QNHLMSFVRSAKQVT	0	84	–	0	
SNRNLSEHP	0	89	0	0	
HKTHEAD	0	49	0	0	

Discussion

T-cell responses known to provide long-term immunity by promoting the formation of memory T cells can quickly respond to future infections by the same pathogen. TCEs can target multiple epitopes across different viral proteins, potentially leading to broader immune protection compared to single B-cell epitope vaccines. Additionally, vaccines based on TCEs designed to target conserved regions of viral proteins reduce the likelihood of immune escape due to viral mutations. The study presents a novel ML-based approach for predicting TCEs with high accuracy, targeting the SARS-CoV-2 pathogen. This approach, utilizing the XGBoost algorithm, demonstrates significant potential in the field of EBV design, offering several advantages over traditional methods. One of the most notable findings is the ability of the proposed model to predict TCEs with a high degree of accuracy, as evidenced by its superior performance compared to other state-of-the-art ML classifiers. The XGBoost model’s accuracy underscores the effectiveness of this approach in identifying potential vaccine targets, particularly in a rapidly evolving virus like SARS-CoV-2. The importance of this achievement is further emphasized by the model’s capability to handle variable-length epitopes, a significant improvement over existing tools like CTLpred, which is limited to 9-mers. The use of physicochemical properties as features in the model also plays a crucial role in its predictive power. The study highlights the significance of features such as the Aliphatic Index, Boman Index, and hydrophobicity, which have been shown to influence peptide immunogenicity. By selecting these features using the Boruta Algorithm, we ensured that the model focused on the most relevant characteristics, enhancing its prediction accuracy. This feature selection process is vital for understanding the underlying molecular mechanisms that contribute to TCE recognition, offering valuable insights for biologists and immunologists. In comparing our approach to existing tools such as NetMHC and CTLpred, we have demonstrated that the proposed model not only addresses the limitations of these methods but also offers a more comprehensive solution for TCE prediction. NetMHC, while effective at estimating peptide-binding capacity, does not directly predict SARS-CoV-2 epitopes. CTLpred, though capable of direct prediction, is restricted by the length of peptide sequences it can analyze. Our model overcomes these limitations by accurately predicting epitopes of varying lengths, providing a more versatile tool for vaccine development. The decision to use XGBoost is that it offers a balance between performance and efficiency, making it a more suitable choice for this study. The model’s scalability and computational efficiency are particularly advantageous in the context of SARS-CoV-2, where rapid and accurate predictions are essential. Furthermore, this study opens up new avenues for future research. The findings suggest that integrating additional features, such as structural properties of peptides, could further enhance the model’s accuracy and applicability. Exploring the combination of TCE-based vaccines with other approaches, such as B-cell or multi-epitope vaccines, could lead to the development of more effective and comprehensive immunization strategies against SARS-CoV-2 and other pathogens. By addressing the limitations of existing tools and providing a robust and efficient method for TCE prediction, this study contributes valuable insights to the ongoing efforts in vaccine development. The implications of this work extend beyond SARS-CoV-2, offering potential applications in the design of vaccines for a wide range of infectious diseases. Although TCE-based EBV design offers significant advantages, it also presents some limitations. Currently, there is less clinical data available on TCE-based vaccines compared to traditional approaches. Further research is needed to validate their efficacy and safety in human trials.

Conclusion

As the COVID-19 pandemic unfolded, marked by genetic variations in the virus over time, the imperative for developing efficacious vaccines became increasingly significant (Niessl, Sekine & Buggert, 2021). While various vaccines have been developed, EBVs have received less attention despite their potential to enhance safety, immunogenicity, and cost-effectiveness (Seder, Darrah & Roederer, 2008). This study presents a unique ML-based computational model that predicts TCEs of the SARS-CoV-2 virus, offering a promising approach to identifying potential vaccine targets thus reducing the time and cost associated with laboratory-based approaches. The performance of the proposed model was evaluated against existing classification models, and it was found to achieve higher accuracy, precision, recall, F score and AUC of 97.6%, 0.963, 0.971, 0.958, 0.99 respectively, on the test set. The results demonstrate that the XGBoost model outperforms existing classification models. Furthermore, the model’s consistency and reliability were evaluated through 5FCV, and it was observed that the proposed model exhibits a practically linear performance with a mean accuracy of 97.58%.While the predictions made by the model will still require experimental validation both in vivo and in vitro, the potential impact of this method is significant in preventing future outbreaks, reducing the virus’s ability to evade immunity, and saving lives. The implications of the findings are numerous like prioritizing epitopes computationally; researchers can streamline the vaccine development process, minimizing the need for extensive experimental validation and reducing costs. Predicting TCE using ML can guide the selection of multiple epitopes for inclusion in a single vaccine formulation, aiming to induce broader and more durable immune responses. In future research, the focus will be on exploring additional physicochemical properties and creating models that are more robust by utilizing advanced ML classifiers to improve classification accuracy. However, it’s important to note that EBVs have not yet received approval for human use due to the fact that peptides, which are their main component, are not highly effective at stimulating an immune response on their own. As a result, adjuvants are typically needed to enhance their immunogenicity (Coffman, Sher & Seder, 2010). This, in turn, allows for smaller doses of the antigen to be used and reduces the number of immunizations required.

Supplemental Information

Supplemental Information 1 Appendix: Grid search results.

Supplemental Information 2 Dataset.

Supplemental Information 3 Peptide sequneces dataset.

Supplemental Information 4 Logistic Regression_Implementation.

Supplemental Information 5 Random Forest Implementation in R programming.

Supplemental Information 6 Neural Network Implementation in R programming.

Supplemental Information 7 SVM Implementation in R programming.

Supplemental Information 8 XGBoost model Implementation in R programming.

Supplemental Information 9 Feature extraction code in R programming.

Supplemental Information 10 Decision Tree Implementation in R programming.

Supplemental Information 11 Instructions.

Supplemental Information 12 Grid search hyperparameter tuning algorithm.

We want to extend our thanks to our biotechnologist friend Dr. Nadiem Nazir with expertise in computational biology and biotechnology for his generous assistance.

Additional Information and Declarations

Competing Interests

Author Contributions

Data Availability

The authors declare that they have no competing interests.

Syed Nisar Hussain Bukhari conceived and designed the experiments, performed the experiments, analyzed the data, performed the computation work, prepared figures and/or tables, authored or reviewed drafts of the article, and approved the final draft.

Kingsley A. Ogudo analyzed the data, prepared figures and/or tables, authored or reviewed drafts of the article, and approved the final draft.

The following information was supplied regarding data availability:

The data is available in the Supplemental File.

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
