# Peer review of "Prediction of antigenic peptides of SARS- CoV-2 pathogen using machine learning"

_PeerJ Computer Science, doi:10.7717/peerj-cs.2319_

## Round 0.1 · original submission · Major Revisions

Reviewers have addressed several major concerns, especially for the method section regarding details presentation and reproductive purposes. These comments are attached below. Please let us know if you have any questions.

Reviewer 1 ·

Basic reporting

The paper presents a study on using machine learning techniques to identify T-cell epitopes (antigenic peptides) that could serve as potential targets for epitope-based vaccine (EBV) design against SARS-CoV-2, the virus responsible for COVID-19.

1. Minor grammatical issues should be addressed to enhance clarity.
2. The introduction could benefit from a clearer statement of the novel contribution of the study, particularly how it advances the current understanding of epitope prediction using machine learning.
3. The literature review includes only publication as late as 2022. It is recommended that the authors conduct a more in-depth review of more recent publications and methodologies.
4. Clarifying how this model differs or improves upon existing models in terms of prediction capabilities or computational efficiency would be beneficial.

Experimental design

1. The methods are described with a commendable level of detail, particularly regarding data handling, feature extraction, and the specifics of the machine learning algorithms. This detailed methodological description supports the potential replicability of the study.
2. The manuscript does not explicitly state how it addresses a specific knowledge gap in the existing scientific literature. While the introduction hints at the advancement over traditional methods, a clearer and more detailed explanation of how this study's approach is innovative or superior to existing models would significantly enhance its contribution to the field.

Validity of the findings

1. The paper would benefit from discussing the potential for model overfitting given the exceptionally high accuracy rates reported and detail any steps taken to mitigate this (besides cross validation).
2. To further strengthen the conclusions, discussions on the implications of these findings for vaccine development and future research directions could be expanded.

Additional comments

Minor Comments:

1. The equation on line 194 is not formatted well.
2. On line 230, yi should be y_i
3. On line 266, what is [38]? On line 119, what is [28]?
4. It is suggested to present the results of grid search in hyperparameter in appendix, in a table or figure.
5. Are the results in Table 3 the same as Figure 5? If so, it is recommended to only keep one, as they share duplicated information.

Cite this review as

Reviewer 2 ·

Basic reporting

Bukhari & Ogudo proposed using machine learning (ML) approaches to predict T-cell epitopes (TCE) for antigenic peptides as candidate vaccine for SARS-Cov-2 pathogen. Using a total ~18k sequences collected from Immune Epitope Database, their analysis benchmarked five state-of-art ML classifiers and observed XGBoost approach achieved highest prediction accuracy (97.6%) in identifying a peptide sequence as epitope in holdout dataset. Their work provides very intriguing results showcasing that ML approaches can be extremely powerful and accurate in nominating candidate epitopes for vaccine development. The manuscript is written well, and results presented clearly.

Experimental design

Since the performance of the other four ML classifiers is close to XGBoost, can the authors add standard logistic regression as a baseline model to compare against?

Validity of the findings

1. Given the impressive high accuracy of XGBoost result, can the authors add feature importance to interpret which feature(s) drive this high accuracy of prediction? This will be very informative for biologist to understand the underlying molecular mechanisms.

2. Can the authors provide an explanation on “linear performance” mentioned in the abstract and in line 441?

Cite this review as

Reviewer 3 ·

Basic reporting

Prediction of antigenic peptides of SARS- CoV-2 pathogen using machine learning by Bukhari et al. delves into identifying potential T-cell epitopes (TCE) for epitope-based vaccine (EBV) design using an in-silico machine learning (ML) paradigm. For this, the authors conducted a workflow that involved querying the Immune Epitope Database, extracting and selecting peptide physiochemical features, and then building the model using XGBoost. The authors also compared their model against other supervised ML methods like support vector machines, decision trees, random forest, and neural networks. The study is noteworthy for its efforts, but it requires more comparative analysis to demonstrate how their method performs against the tools already available and the existing ML methods used in predicting SARS-CoV-2 TCE, as listed by the corresponding author of this manuscript in a separate review article with doi: 10.3390/pathogens11020146.

Minor comments -
1. Rather than use adjectives like "large," it would be more informative if the authors could specify the size of the CoV genome [line 48]

2. Several receptors and co-receptors are involved in cell infection with SARS-CoV-2 doi: 10.1134/S1990747822060034. Why do the authors only mention ACE2? [line 59]

3. The authors could consider providing the length of the predicted TCEs to ensure readers have a comprehensive understanding.

4. The authors provide performance metrics such as accuracy, prediction, and AUC when comparing XGBoost with other machine learning methods. It would be helpful if the authors also included a table comparing various aspects such as dataset size requirement, training time, hyperparameters, robustness, and overfitting.

Experimental design

Major concerns -
1. One central aspect that could be further elaborated is the authors' choice of XGBoost over deep learning methods, primarily when the latter is more frequently used in epitope prediction in general. It would be beneficial if the authors could provide some context on this in the introduction or discussion, as it would help to understand the rationale behind their decision.

2. The authors' exclusive focus on TCE-based EBV design for SARS-CoV-2 is an interesting choice. To provide a more comprehensive understanding, it would be beneficial for the authors to clarify the potential advantages of this approach over B-cell epitope or multi-epitope vaccine design in their discussion section. They could also compare it to other relevant papers, such as doi:10.1038/s41598-021-81749-9, while addressing any potential shortcomings.

3. In the literature review section of the manuscript, the authors discuss synthesizing epitopes using cytotoxic T lymphocytes (CTLs) and helper T lymphocytes (HTLs) of SARS-CoV-2 proteins. However, their methodology and other sections of the paper refer to the prediction of TCEs in general without specifying if they are focusing on CTLs, HTLs, or both. If they used one over the other, they should provide a reason for their choice. If both types were involved, it would be worthwhile to give a breakdown of XGBoost's prediction accuracy and compare it against other ML algorithms. This is especially important since the prediction of CTL and HTL epitopes involves different methodologies due to their distinct pathways of antigen processing and presentation.

4. The authors must necessarily compare their "unique" ML-based computational model with existing tools or other ML methods used in the field. This is warranted as it will help the reader understand the strengths and weaknesses of the current method compared to others that have been published.

Validity of the findings

please see previous section

Additional comments

none

Cite this review as

---

## Round 0.2 · Minor Revisions

Thank you so much for your responses to the comments. One of the reviewers still wants a small modification to the discussion.

Reviewer 1 ·

Basic reporting

I appreciate the authors for revising the manuscript, and writing a clear and detailed response letter. The revision has addressed my previous comments.

Experimental design

no comment

Validity of the findings

no comment

Additional comments

no comment

Cite this review as

Reviewer 2 ·

Basic reporting

None

Experimental design

None

Validity of the findings

None

Cite this review as

Reviewer 3 ·

Basic reporting

The authors have addressed all my comments.
I would urge them to expand on the discussion section and make it a section of its own than merge it with results.

Experimental design

-

Validity of the findings

-

Cite this review as

---

## Round 0.3 · accepted · Accept

Thank you so much for your responses to the reviewers' comments. We do not have additional comments or concerns.